

# 1 Quantifying the Impacts of Marine Aerosols over the Southeast 2 Atlantic Ocean using a chemical transport model: Implications 3 for aerosol-cloud interactions

Mashiat Hossain[1], Rebecca M. Garland[2], Hannah M. Horowitz[1]
[1] Civil and Environmental Engineering, University of Illinois Urbana-Champaign, IL, USA
[2] Department of Geography, Geoinformatics and Meteorology, University of Pretoria, Pretoria, South Africa
*Correspondence to*: Mashiat Hossain (mashiat3@illinois.edu), Hannah M. Horowitz (hmhorow@illinois.edu)
**Abstract.** The southeast Atlantic region, characterized by persistent stratocumulus clouds, has one of the highest
uncertainties in aerosol radiative forcing and significant variability across climate models. In this study, we analyze
the seasonally varying role of marine aerosol sources and identify key uncertainties in aerosol composition at cloud-
relevant altitudes over the southeast Atlantic using the GEOS-Chem chemical transport model. We evaluate simulated
aerosol optical depth (AOD) and speciated aerosol concentrations against those collected from ground observations
and aircraft campaigns such as LASIC, ORACLES, and CLARIFY, conducted during 2017. The model consistently
underestimates AOD relative to AERONET, particularly at remote locations like Ascension Island. However, when
compared with aerosol mass concentrations from aircraft campaigns during the biomass burning period, it performs
adequately at cloud-relevant altitudes, with a normalized mean bias (NMB) between -3.5% (CLARIFY) and -7.5%
(ORACLES). At these altitudes, organic aerosols (63%) dominate during the biomass burning period, while sulfate
(41%) prevails during austral summer, when dimethylsulfide (DMS) emissions peak in the model. Our findings
indicate that marine sulfate can account for up to 69% of total sulfate during high DMS period. Sensitivity analyses
indicate that refining DMS emissions and oxidation chemistry may increase sulfate aerosol produced from marine
sources, highlighting their overall importance. Additionally, we find marine primary organic aerosol emissions may
substantially increase total organic aerosol concentrations, particularly during austral summer. This study underscores
the imperative need to refine marine emissions and their chemical transformations to better predict aerosol-cloud
interactions and reduce uncertainties in aerosol radiative forcing over the southeast Atlantic.

## 25 1 Introduction

Marine aerosols are a primary contributor to natural atmospheric aerosols, and consequently influence the Earth's
radiative balance (Spracklen et al., 2008; Vignati et al., 2001). Aerosols in the marine boundary layer have significant
impact on the properties of low-altitude marine clouds, particularly their ability to reflect solar radiation and cool the
climate (Seinfeld and Pandis, 2016; Wood, 2012; Chen et al., 2014; Quinn et al., 2017). The southeast Atlantic (SEA)
is marked by a persistent deck of low-level stratocumulus (Sc) clouds. However, aerosol radiative forcing in the region
exhibits highest uncertainty and one of the largest intermodel spread, primarily due to the differences in modeled cloud
fraction (Stier et al., 2013; Zuidema et al., 2016), as well as aerosol and cloud properties (Doherty et al., 2022). These
uncertainties are further compounded by poorly constrained optical properties of the absorbing biomass burning



aerosols, the vertical distribution of aerosols relative to these clouds and the interaction of aerosols with marine
boundary layer clouds (Zuidema et al., 2016), and limited observations of aerosols and their precursors in remote
marine environments (Croft et al., 2021). In this study, we investigate the role of marine aerosols and sources of
uncertainty affecting aerosol composition in this critical region of aerosol-cloud interactions over the SEA.
The SEA region encompasses the Benguela upwelling system (BUS), renowned for its high primary production of
marine phytoplankton and fish populations (Shannon and Nelson, 1996; Jarre et al., 2015). This elevated
phytoplankton activity serves as the main natural source of the volatile organic compound dimethylsulfide (DMS),
thereby influencing the global tropospheric sulfur budget (Andreae, 1990; Bates et al., 1992). Once released into the
atmosphere through air-sea exchange, DMS undergoes complex chemical transformations. In the gas phase, it is
oxidized to form $H_2SO_4$ and methanesulfonic acid (MSA), which has implications for new particle formation (Chen
et al., 2015); while in the aqueous phase, it leads to the production of MSA and sulfate aerosols, impacting cloud
microphysical properties (Kaufman and Tanré, 1994). Despite its significance, the exact mechanisms of DMS
oxidation and subsequent formation of sulfate and MSA remain inadequately understood (Ravishankara et al., 1997;
Barnes et al., 2006; Hoffmann et al., 2016), leading to largest uncertainty of aerosol radiative forcing within climate
models (Carslaw et al., 2013). Additionally, marine aerosols comprise primary aerosols such as sea spray aerosols,
which consist of sea salt and organic matter, released into the atmosphere primarily by the bubble-bursting process
(O'Dowd and De Leeuw, 2007; Russell et al., 2010; Prather et al., 2013; Brooks and Thornton, 2018). Investigating
the uncertainties related to marine emissions and chemistry are crucial to refine our understanding of the impacts of
marine aerosols on climate.
The SEA lies at the confluence of not only marine aerosols, but other natural and anthropogenic aerosols from local
and distant origin (Andreae et al., 1995; Swap et al., 1996; Formenti et al., 1999; Swap et al., 2003; Tournadre, 2014).
During the austral spring (August to October), seasonal fires in the neighboring southern African region contribute
nearly one-third of global total biomass burning emissions (van der Werf et al., 2010). This seasonal influx of biomass
burning aerosols aloft interacts with the underlying Sc deck, introducing considerable variability into aerosol forcing
assessments in the SEA region (Lindesay et al., 1996; Swap et al., 2003). To address these uncertainties, several
international field campaigns were conducted between 1992 and 2018 during the peak biomass burning season (Swap
et al., 2003; Formenti et al., 2019; Haywood et al., 2021; Redemann et al., 2021). Despite the region being a prolific
source of marine aerosols throughout the year, the potential impact of aerosols on regional climate dynamics through
interactions with the persistent low-level marine clouds outside of the biomass burning season has been largely
overlooked.
Here, we use the GEOS-Chem global chemical transport model to analyze high-resolution, seasonally varying aerosol
composition at the altitudes of persistent stratocumulus clouds over the SEA. We specifically focus on the role of
marine aerosols, analyzing their contributions to sulfate and organic aerosol concentrations. We evaluate simulated
aerosol optical depth (AOD) and speciated aerosol concentrations against observational data from the Aerosol Robotic
Network (AERONET) and the Layered Atlantic Smoke Interactions with Clouds (LASIC), ObseRvations of Aerosols



above CLouds and their intEractionS (ORACLES), and CLoud–Aerosol–Radiation Interaction and Forcing
(CLARIFY) field campaigns during the year 2017. We assess the sensitivity of our results to uncertainty in DMS
oxidation mechanisms and emissions of DMS, $SO_2$, and marine primary organics. Our findings aim to enhance
understanding of the seasonally varying role of marine aerosols in aerosol-cloud interactions in the SEA by a
comprehensive evaluation of aerosol composition at cloud altitudes.

## 2 Methodology

### 2.1 Model Description

Here, we use the GEOS-Chem 3D atmospheric chemical transport model version 13.3.3 with detailed gas- and aerosol-
phase tropospheric chemistry (https://zenodo.org/records/5748260). The model is driven by meteorology from the
Modern-Era Retrospective Analysis for Research and Applications, Version-2 (MERRA2) reanalysis, from the NASA
Global Modeling Assimilation Office (GMAO) (Gelaro et al., 2017). We perform nested grid simulations over the
southwestern coast of Africa (40°W-20°E, 0-40°S) with a horizontal resolution of 0.5° by 0.625° and extending over
47 vertical layers from the surface to 0.01hPa. A chemical time step of 20 minutes and transport time step as 10
minutes is applied, as recommended by Philip et al. (2016). Prior to the target year, 2017, we conduct a 6-month spin-
up simulation. Boundary conditions are obtained from global simulations performed at 4° latitude × 5° longitude
horizontal resolution for the same year after a 6-month initialization.
In GEOS-Chem, carbonaceous aerosol includes organic aerosols (OA) and black carbon (BC). BC follows Park et al.
(2003) and Wang et al. (2014). Organic aerosol follows the "simple" scheme which treats primary organic aerosol
(POA) as non-volatile and includes irreversible direct yield of SOA from precursors (Pai et al., 2020). Sulfate
(Alexander et al., 2009), nitrate (Jaeglé et al., 2018), and ammonium (Fountoukis and Nenes, 2007) thermodynamic
partitioning is estimated using the ISORROPIA II thermodynamic model (Fountoukis and Nenes, 2007). Monthly
anthropogenic emissions follow the Community Emissions Data System (CEDSv2) inventory (Hoesly et al., 2018).
Biomass burning emissions are calculated using the Global Fire Emissions Database (GFED4.1s) at 0.25°×0.25°
spatial resolution, with fractional daily and 3-hourly scaling factors applied to the cumulative monthly data (van der
Werf et al., 2017). DMS emissions in the standard model use the Lana et al. (2011) climatology, which compiles DMS
concentrations using data from the Global Surface Seawater DMS Database (http://saga.pmel.noaa.gov/dms/)
collected from 1972 to 2009, incorporated with additional observations from the South Pacific (Lee et al., 2010). The
standard DMS oxidation mechanism in the model includes only three gas-phase DMS reactions, which directly yield
$SO_2$ and MSA according to the reaction mechanism outlined by Chin et al. (1996), and incorporates updated reaction
rate coefficients from Burkholder et al. (2015). Sea-salt aerosol (SSA) emissions from the open ocean are sea surface
temperature-dependent (Jaeglé et al., 2011). Dust emissions include natural dust (Fairlie et al., 2007) and
anthropogenic dust from the AFCID inventory (Philip et al., 2017).
In this study, we carry out multiple simulations to explore the sensitivity of marine aerosols to various emission
sources. To quantify the impact of marine sources on sulfate aerosols within the stratocumulus cloud layer, we perform



a high-resolution (0.5° x 0.625°) marine emissions only sensitivity simulation where $SO_2$ and $SO_4$ emissions from
anthropogenic sources, biomass burning, volcanic activity, ships and aviation were turned off. Additionally, to
investigate the sensitivity of DMS emission fluxes to surface ocean DMS concentrations, we perform an additional
simulation with DMS concentrations from Galí et al. (2018). In this dataset, DMS concentrations are estimated through
a remote-sensing algorithm that integrates satellite-derived estimates of chlorophyll and light penetration, along with
climatological mixed layer depth (Galí et al., 2018). Furthermore, we assess the impact of adding marine POA, co-
emitted with sea-salt aerosols (Gantt et al., 2015), on the overall organic aerosol burden, which is not included in the
standard model configuration. Finally, to evaluate how uncertainty in biomass burning $SO_2$ emissions affects the
relative importance of marine emissions to sulfate aerosol, we conduct two sets of sensitivity simulations using the
Quick Fire Emissions Dataset (QFED) (Darmenov & da Silva, 2013; Das et al., 2017), and the Global Fire
Assimilation System (GFAS) (Kaiser et al., 2012; Su et al., 2023). Each of these inventories differ in data sources,
methodology, temporal resolution and plume injection height. These sensitivity analyses were conducted for the year
2017, following a six-month spin-up period. Details regarding the spatial resolution used in each sensitivity analysis
are provided in Table A1.
**2.2 Ground-based measurements**
We evaluate simulated aerosol optical depth (AOD) against AOD retrieved from the ground-based Aerosol Robotic
Network (AERONET) of sun photometers with direct sun measurements every 15 min (Holben et al., 1998). We use
Level 2.0 Version 3 data that have improved cloud screening algorithms (Giles et al., 2019). We strategically select
nine sites in the study domain along coastal and oceanic regions, as shown in Fig. 1. Site information, including the
coordinates, number of months with available data and the average daily AOD, is summarized in Table A2. The
AERONET monthly average AOD is computed from daily averages for sites with at least 3 months of observations
during the model simulation period (year 2017) and months with at least 15 days of measurements. These are then
compared with the monthly mean AOD from the GEOS-Chem model.
The modeled AOD is computed at 550 nm wavelength by vertically integrating scattering and absorption coefficients
based on the properties of various aerosol components, such as size distributions, hygroscopicity, refractive indices,
and densities (Latimer and Martin, 2019). For comparison with modeled monthly AOD, daily measurements at each
site at 440 nm are first interpolated to the standard wavelength of 550 nm using the local Ångström exponent between
440 and 870 nm channels, following the Ångström power law (Eq. (1); Martínez-Lozano et al., 1998). These
interpolated values are then averaged to calculate the observed mean monthly AOD. The interpolation formula used
is:
$AOD_{(550nm)} = AOD_{(440nm)} * \left(\frac{550}{440}\right)^{-\alpha ext\left(\frac{440}{870}\right)}$                                (1)





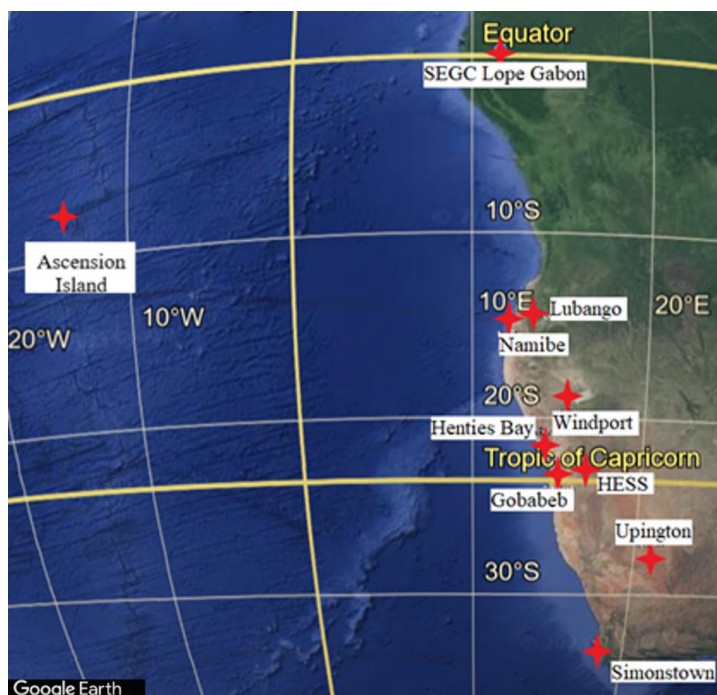

**Figure 1: Map of AERONET sites used for model evaluation (© Google Earth).**

In addition, we evaluate the model's relative aerosol composition against measurements from the Atmospheric Radiation Measurement (ARM) facility on Ascension Island during the LASIC campaign, conducted from January to October 2017. LASIC employed an Aerodyne aerosol chemical speciation monitor (ACSM) to quantify sulfate, nitrate, ammonium, and organic aerosol mass concentrations. Barrett et al. (2022) reported that aerosol mass concentrations of individual components observed by the LASIC ACSM were 2 to 4.5 times lower than those measured by the aerosol mass spectrometer (AMS) aboard the CLARIFY campaign aircraft. Hence, we evaluate the relative rather than absolute aerosol speciation in GEOS-Chem against the LASIC ACSM.

**2.3 Aircraft measurements**

We evaluate simulated aerosol composition against airborne measurements from two campaigns, NASA ORACLES (Redemann et al., 2021; Ryoo et al., 2021) and UK CLARIFY (Haywood et al., 2021). The ORACLES field campaign used the NASA P-3 aircraft to make measurements based out of São Tomé and Príncipe while CLARIFY used the FAAM BAe-146 aircraft around Ascension Island for data collection. The ORACLES aircraft primarily conducted morning sampling, between 8:00-13:00 UTC, while the CLARIFY aircraft often sampled extended hours, typically from 7:00-18:00 UTC. Both campaigns occurred during the austral winter/spring (August-September), corresponding with peak biomass burning events in southern Africa (Adebiyi et al., 2015). Figure 2 shows the flight tracks for these campaigns. The primary instruments and references for each campaign are listed in Table 1.



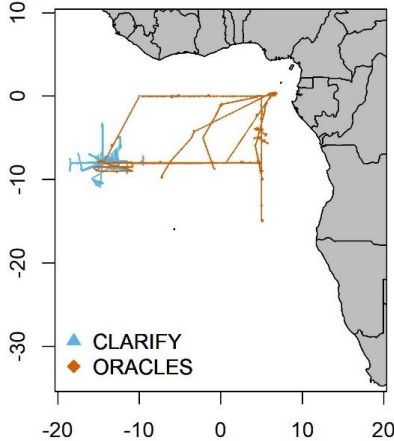


**Figure 2: Flight tracks from the two aircraft campaigns used to evaluate the model, CLARIFY (in blue) and ORACLES (in orange), conducted during August-September 2017 over the southeast Atlantic region.**

To facilitate comparison between airborne measurements and the GEOS-Chem model, we sampled the model to the
nearest grid box, both temporally and spatially, along the flight tracks. Observations from both campaigns are reported
at 1-minute averaging intervals, while the model operates at a 10-minute temporal resolution (see Sect. 2.1). Aerosol
concentrations from the campaigns are reported as mass concentrations at standard temperature and pressure (STP:
273 K, 1 atm). The modeled concentrations are thus also standardized to STP conditions.
**Table 1:** Aircraft campaigns in the southeast Atlantic used for model evaluation during the biomass burning season

| Campaign | Date range (Duration) | Instruments* | Aerodynamic Diameter (µm) | Altitude from surface (km) | Primary Reference |
|---|---|---|---|---|---|
| CLARIFY | 7th August–4th September 2017 (99h) | C-ToF-AMS | 0.05 to 0.60 | 0 to 8 | Haywood et al., 2021 |
| ORACLES | 16th August–6th September 2017 (112h) | HR-ToF-AMS | 0.07 and 0.70 | 0 to 7 | Redemann et al., 2021 |

*Compact Time-of-Flight (C-ToF), High Resolution Time-of-Flight (HR-ToF), Aerosol Mass Spectrometer (AMS)
**3.1 Model Evaluation**
**3.1.1 Seasonal variation of AOD**



The spatial distribution of seasonal mean AOD from GEOS-Chem for the year 2017 is presented in Fig. 3. Three
distinct seasonal periods reflect dominant atmospheric and oceanic processes. These include the high DMS emission
period in the SEA, during the months of January, February, November, and December (JFND); the peak biomass
burning season in southern Africa, spanning from July to October (JASO); and the transitional season, encompassing
March, April, May, and June (MAMJ).

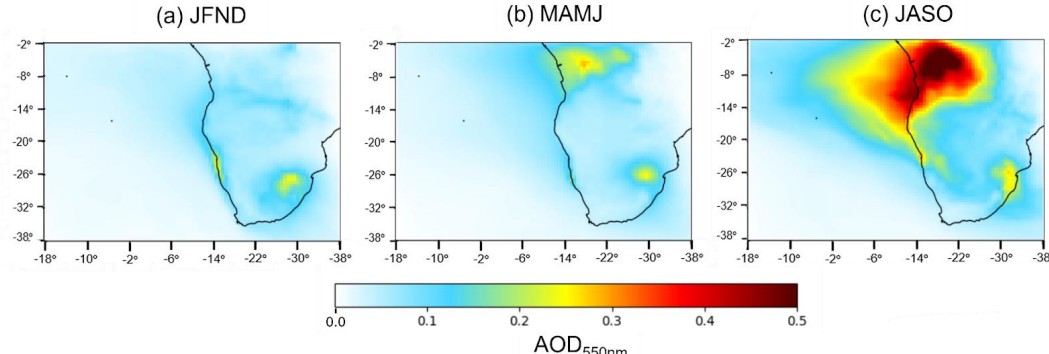


**Figure 3: Spatial distribution of seasonal mean modeled AOD at 550 nm for 2017. Seasons are as follows: (a) the peak DMS**
**emission period (JFND), (b) the transitional period (MAMJ), and (c) the peak biomass burning period (JASO).**
The simulated DMS emissions, based on Lana climatology (2011), indicates that emissions in the BUS region peak
in January, leading to elevated concentrations of sulfate aerosols. This increase, combined with dust emissions from
the Namib desert, contributes to an AOD hotspot as depicted in Fig. 3a on the southwestern coast. In the JASO period
(Fig. 3c), modeled AOD increases due to biomass burning aerosols, originating from savannah fires in Central and
southern Africa and transported westward towards the SEA region by the southern African easterly jet (Adebiyi and
Zuidema, 2016). The spatial distribution of mean transitional period AOD (Fig. 3b) features hotspots in Namibia and
southern Africa, which coincide with dominant anthropogenic sources and the onset of biomass burning in Central
Africa.





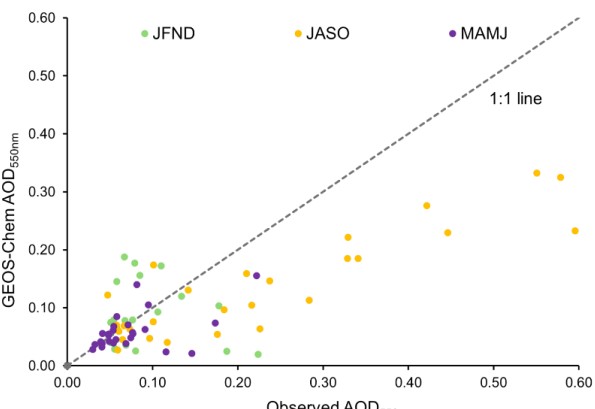


**Figure 4: Modeled AOD$_{550nm}$ (Y-axis) with respect to AERONET AOD$_{550nm}$ (X-axis). Each data point represents the monthly mean values for each station color-coded by season (green- DMS period, yellow - biomass burning period, purple-transitional period). The dotted line depicts the 1:1 relationship.**

Figure 4 shows the correlation of monthly average AERONET and GEOS-Chem AOD across the nine selected sites (see Sect. 2.1 and Fig. 1), with the three seasonal periods indicated by color (green for peak DMS emission season-JFND, yellow for biomass burning season-JASO, and purple for the transition period-MAMJ). Table 2 compiles the performance of monthly mean GEOS-Chem AOD with respect to AERONET AOD by season. JASO exhibits the strongest correlation (R = 0.92), which is statistically significant (p <0.05). The transitional period (MAMJ) shows a moderate correlation (R = 0.51) with a normalized mean bias (NMB) of -9.5%. A notably low correlation coefficient (R = -0.12) with a positive bias (12.5%) is seen during the summer period (JFND), predominantly due to anomalies at two sites. This period witnesses a considerable underestimation of AOD at Ascension Island, alongside an overestimation of dust aerosol at Gobabeb. Excluding these two sites from the analysis, both the correlation coefficient and NMB improve to 0.61 (p = 0.55) and -8% respectively, indicating better model performance at the remaining 7 sites. This underestimate of AOD at Ascension Island (Fig. A1 in the Appendix) during summer (JFND) suggests potential model limitations in accurately simulating natural aerosol emissions such as sea salt and marine biogenic emissions. Meanwhile, the AOD discrepancy at Ascension Island in the biomass burning season, may be due to the underestimate of transatlantic transport of light-absorbing carbon aerosols (Das et al., 2017) and deviations in its spatial distribution from typical zonal patterns over the Atlantic (Adebiyi et al., 2023). Furthermore, Table 2 shows that the model underestimates AOD during JASO by 26.5% (NMB) across the domain during. This underestimate may stem from the model's bulk aerosol scheme which inadequately captures the optical properties of aerosols and is compounded by a low relative humidity bias (Zhai et al., 2021). The bulk scheme also assumes all aerosols are externally mixed, which contrasts with the variable degree of particle mixing states in the atmosphere (Yu et al., 2012). Additionally, studies like Hodzic et al. (2020) using NASA ATom aircraft data, indicate that GEOS-Chem substantially underestimates oxidation levels of organic aerosols in remote areas, which could affect estimates of their burden and optical properties.



Table 2: Statistical parameters of monthly mean modeled AOD with respect to observed AOD at the AERONET sites by season

| Time period | Number of observations | Correlation coefficient (R) | Normalized mean bias (NMB) (%) | Root-mean square error (RMSE) |
|---|---|---|---|---|
| JFND | 20 | -0.12 (p = 0.75) | 12.5 | 0.079 |
| MAMJ | 26 | 0.51 (p = 0.15) | -9.5 | 0.044 |
| JASO | 28 | 0.92 (p = 0.018) | -26.5 | 0.15 |

We evaluate the relative aerosol speciation simulated at Ascension Island against monthly mean ACSM observations during the LASIC campaign (see Sect. 2.2) available for January–October 2017 (Fig. A2 in the Appendix). The seasonality of the relative contributions of organic aerosols and sulfate are consistent between the model and observations. However, the model underestimates the relative contribution of sulfate during most months, while generally overestimating the proportion of organics. An increase in the transport of biomass burning organic aerosols would further worsen the model underestimate of sulfate. The modeled relative contribution of sulfate is closest to that observed in January and February, when simulated DMS emissions in the region are high (Lana et al., 2011), with a slight overestimate in the latter.

**3.1.2 Vertical profiles of aerosol composition**

Figure 5 depicts the mean vertical profiles of speciated aerosol mass concentrations observed during ORACLES and CLARIFY aircraft campaigns in August–September 2017 (the biomass burning season), compared to GEOS-Chem (see Sect. 2.2 and Table 1). The cloud top height in the SEA region generally falls between 0 to 2 km (Redemann et al., 2021). Findings from Diamond et al. (2018) indicate that aerosols below clouds in this lower atmospheric layer can also substantially impact cloud microphysics. At these altitudes (0–2 km), GEOS-Chem performs well against AMS measurements of total aerosol mass, which includes sulfate, nitrate, ammonium and organics from these campaigns, with an NMB between -3.5% (CLARIFY) to -7.5% (ORACLES). At mid-altitudes (2–4 km), the model achieves moderate agreement, with NMB values spanning -19% (ORACLES) to -57% (CLARIFY). However, the model demonstrates a pronounced bias at higher altitudes (4–7 km), where NMB values drop to -92% (ORACLES) to -93.5% (CLARIFY), underscoring challenges in accurately modeling aerosol concentrations at these elevations. Pai et al. (2020) suggests that the model underestimation of organic aerosol loading at mid-tropospheric heights is linked to the surface injection treatment of fire emissions in GFED4.1s. Recent studies by Wizenberg et al. (2023) and Marvin et al. (2024) concur that fire injection scheme is a critical source of model uncertainty, emphasizing the potential

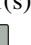


importance of accurate fire injection modeling in the free troposphere. Nonetheless, our study focuses on aerosol
composition within cloud-relevant altitudes to improve our understanding of aerosol-cloud interactions and their
climate implications. The observed vertical distribution of aerosol mass concentrations (left panels of Fig. 5), indicates
that 18% and 36% of the aerosol mass for the ORACLES and CLARIFY campaigns, respectively, is located below 2
km, within columns extending up to flight altitudes of 7 km and 8 km. However, the model simulates elevated aerosol
mass at these lower altitudes, 24% and 50% of the column for ORACLES and CLARIFY, respectively.

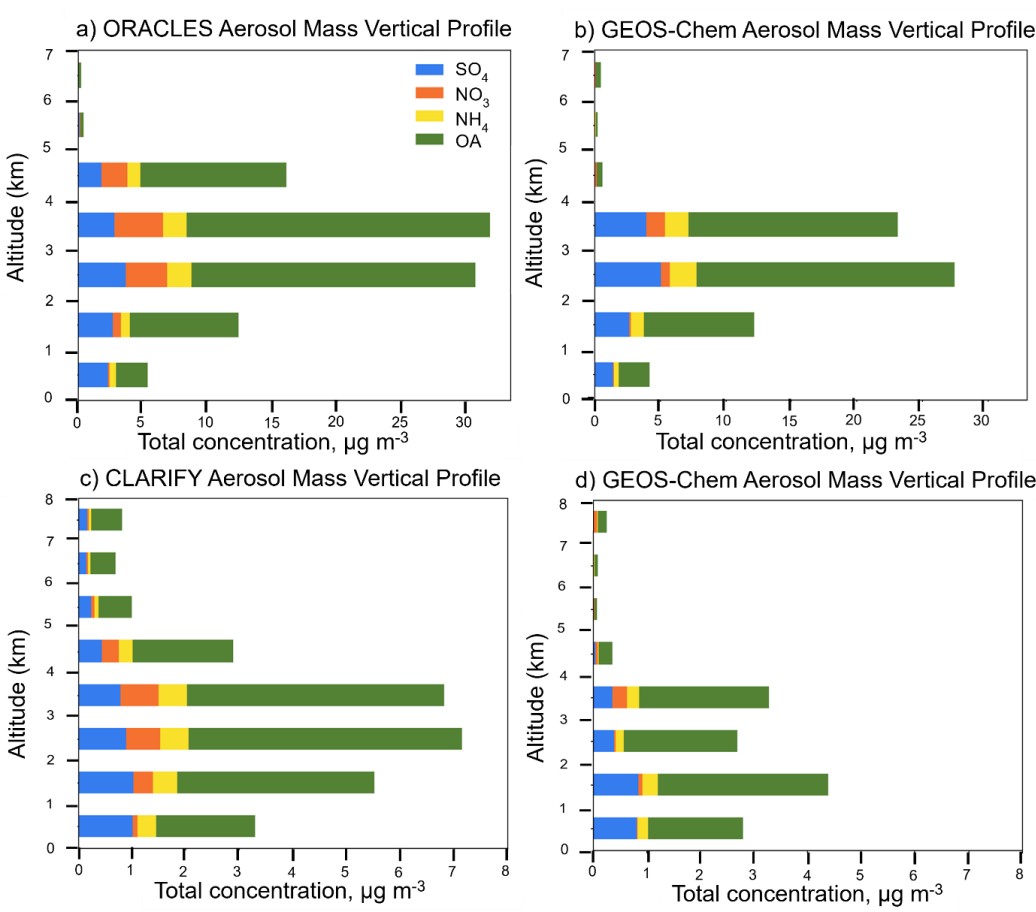


**Figure 5: Average vertical profiles of simulated and observed aerosol mass during August–September 2017 (peak biomass**
**burning season) from aircraft campaigns. The left column presents the vertical distribution of aerosols observed during the**
**ORACLES flight campaign (panel a) and the CLARIFY flight campaign (panel c) at STP (see Sect. 2.3). The right column**
**displays the GEOS-Chem model simulations along the respective flight tracks of each campaign (panels b and d). All data**
**are averaged over 1 km vertical bins.**
At altitudes where clouds persist in the domain (0 to 2 km), sulfate and organic aerosols are the dominant aerosol
types. Here, the model effectively captures the mass concentration of organic aerosols, with an NMB ranging from -





0.40% for ORACLES to -14% for CLARIFY. However, it underestimates sulfate aerosol concentrations by 19% at
cloud altitudes for both campaigns. For other aerosol types and altitudes, the model consistently underestimates
concentrations, except for sulfate and ammonium aerosols between 2 to 4 km during the ORACLES campaign, which
the model overestimates by 40% and 4.6%, respectively. The model captures the total aerosol mass from 0 through 7
km for sulfate and ammonium aerosols during the ORACLES campaign, with only minimal underestimations of 1.5%
and 0.7%, respectively. This indicates a potential discrepancy in the vertical distribution of these aerosols rather than
in total mass.
**3.2 Seasonal variation in aerosol composition and sources at cloud altitudes**

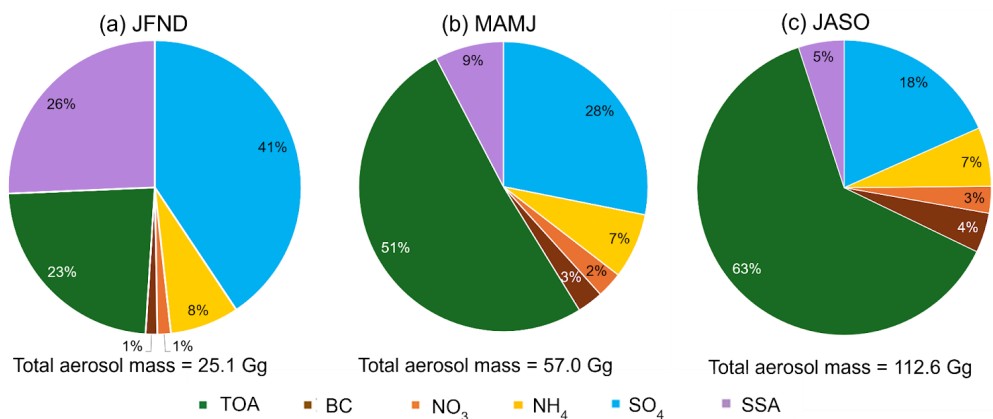


**Figure 6: Simulated mean fractional aerosol composition at cloud heights (0 –2 km) over the ocean in the stratocumulus**
**sub-domain (0–35° S, 20° E–20° W) by season: (a) JFND, (b) MAMJ, and (c) JASO. Here SO₄, NH₄, NIT, BC, TOA, SSA**
**represents sulfate, ammonium, nitrate, black carbon, total organic aerosol and accumulation-mode sea salt aerosols,**
**respectively.**
Figure 6 presents the simulated seasonal mean aerosol fractional composition within cloud-relevant altitudes (0-2 km),
averaged over the ocean only across the subdomain (0–35° S, 20° E – 20° W) (see the map shown in Fig. 7). This area
is strategically selected to coincide with the persistent Sc cloud deck and enhance our analysis of aerosol-cloud
interactions. Organic aerosols, an indicator of biomass burning, predominate during both the biomass burning (JASO)
and transitional (MAMJ) periods. In contrast, sulfate aerosols dominate during austral summer, likely influenced by
the high primary production from coastal upwelling that leads to DMS emissions. We investigate the model
representation of sulfate and these processes further in subsequent sections. An increase in the accumulation-mode
sea-salt aerosols (radius 0.01–0.5 μm) contribution (total mass of 6.7 Gg) is observed in summer (Fig. 6a) as well,
compared to other seasons (5.2 Gg during MAMJ and 5.8 Gg during JASO), owing to the peak wind speeds in the
southern Benguela region in this season (Hutchings et al., 2009). Black carbon, ammonium, and nitrate aerosols make
minor contributions to simulated aerosol mass at cloud height throughout the year.





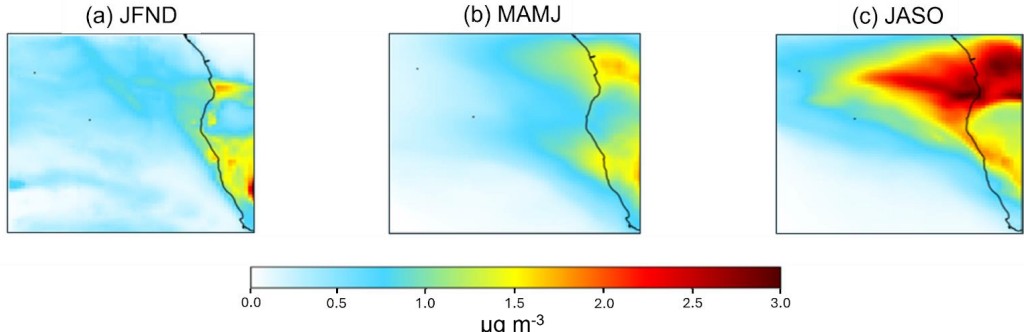


**Figure 7: Spatial distribution of simulated mean sulfate aerosol concentrations averaged over cloud altitudes (0–2 km) in**
**the sub-domain (0–35° S, 20° E–20° W) by season in 2017: (a) peak DMS emission season (JFND), (b) transitional phase**
**(MAMJ), and (c) biomass burning season (JASO).**

### 3.2.1 Drivers of sulfate aerosol and importance of marine precursor emissions

Sulfate aerosols are the most or 2[nd] most important aerosol component in cloud heights over the SEA (Fig. 6). We
examine the sources of sulfur emissions within the model in Figure A3. Within the broader domain (0–40° S, 40° E–
20° W), anthropogenic activities are the largest source of sulfur emissions throughout the year (Fig. A3), followed by
DMS emissions from the ocean. DMS emissions become more pronounced during the austral summer, peaking in
January. Additionally, biomass burning contributes to $SO_2$ emissions seasonally, becoming the 3[rd] most important
source of total sulfur emissions during July - September (Fig. A3).

To improve understanding of the processes driving sulfate aerosol concentrations in the region, we examine its
simulated spatial distribution averaged by season over the cloud height (0–2 km) in Fig. 7. Elevated concentrations of
DMS, resulting from higher rates of primary production (Lana et al., 2011; Galí et al., 2018), lead to an increase in
sulfate concentrations along the coastline of the Benguela region and the inner shelf of Namibia during JNFD (Fig.
7a), aligning with the AOD hotspot observed in Fig. 3a. This is consistent with the simulated dominance of sulfate
aerosols at cloud-relevant altitudes during JFND (Fig. 6a). During the biomass burning months (JASO), while their
relative contribution decreases (Fig. 6c), sulfate aerosols display a pronounced increase in absolute concentration (Fig.
7c) as a consequence of savanna fire emissions from southwestern Africa (van der Werf et al., 2010; Das et al., 2017).
As outlined in the AOD evaluation (Sect. 3.1.1), the model underestimates the transport of emissions to remote sites
(Fig A1), resulting in a steep gradient in sulfate concentrations from the eastern landmass towards the western open
ocean.

To quantitatively estimate the contribution of marine precursor emissions to sulfate aerosols, we compare the sulfate
mass between the standard and marine emissions only sensitivity simulations (Sect. 2.1). Figure 8 shows seasonally
averaged vertical profiles over the ocean region of the Sc sub-domain (0°–35° S, 20° E–20° W). The figure presents
the marine-only sulfate mass and the total sulfate mass from the standard simulation (left panels), and the ratio of
marine sulfate to total sulfate (right panels). Vertical profiles were computed by summing the sulfate mass within each



296    grid box, scaled by the grid box ocean fraction, across latitude and longitude within each vertical layer of the model,

297    and then averaged temporally across each season.

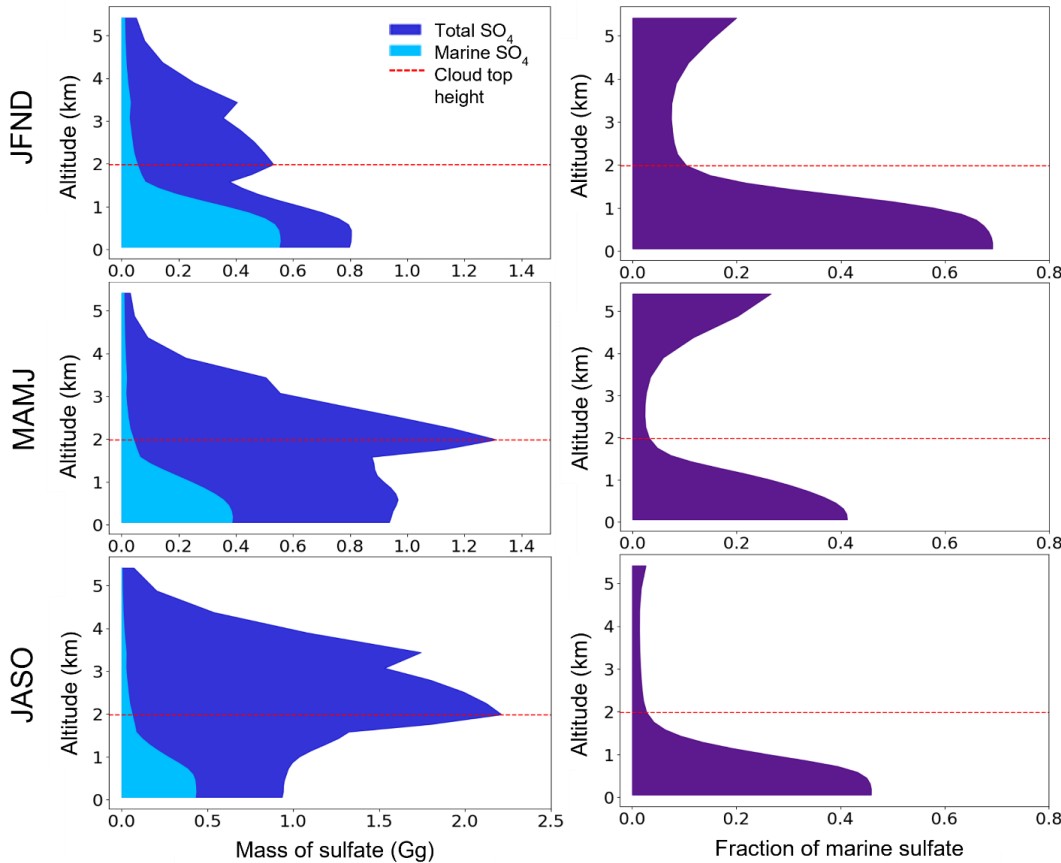

298

**Figure 8: Simulated vertical profiles of sulfate aerosol mass over oceanic regions within the sub-domain (0°–35° S, 20° E–20° W) by season. The left panel shows the mass of total and marine sulfate aerosols, and the right panel indicates the sulfate fraction from marine sources. The top row corresponds to the peak dimethyl sulfide (DMS) emission period (JFND); the middle row to the transitional period (MAMJ); and the bottom row to the peak biomass burning period (JASO) (note: the bottom left panel displays a higher x-axis scale). The upper red dashed line denotes the typical maximum cloud top height (Redemann et al., 2021).**

Our analysis highlights the substantial influence of marine sulfur sources on sulfate during JFND, as evidenced in the
top left panel of Fig. 8. During this period the proportion of marine sulfate reaches up to 69.1% within-cloud (from
surface to 2 km). The contribution of marine sulfate within the cloud in the subsequent periods is reduced (ranging
between 2.7–45.9%; Fig. 8). We find that marine-sourced sulfate mass remains fairly consistent throughout the year
(Fig. 8, left panels), with variations in the marine sulfate fraction (Fig. 8, right panels) mainly due to changes in land-
based sulfate sources. Total sulfate mass during seasons influenced by biomass burning (MAMJ and JASO) peaks at





2 km, with greater mass above 2 km during peak biomass burning (JASO) in contrast to JFND where mass peaks
within clouds (0–2 km).
Table A3 summarizes the monthly mean percent contribution of marine sulfate averaged across cloud altitudes (0–2
km). The annual average total sulfate mass and marine sulfate mass is 16.2 Gg and 3.5 Gg, respectively. The within-
cloud marine sulfate contribution peaks in January (57.7%) and is smallest in September (10.3%). Thus, our analysis
suggests that DMS emissions influence sulfate aerosol formation and their interactions with clouds in the region
throughout most of the year, excepting only the peak biomass burning season. This emphasizes that constraining
marine sulfur sources and chemistry both in chemical transport and climate models may improve representation of
aerosol-climate dynamics in the SEA region. Limited available observations suggest the model is biased low in AOD
throughout most of the year (Sect. 3.1.1), and underestimates sulfate aerosol concentrations in August and September
at cloud altitudes (Sect. 3.1.2, Fig. 5). We explore related uncertainties and their implications in the following
sections.

### 3.3 Uncertainties

#### 3.3.1 Assessing variations in DMS emission rates and oxidation mechanism on sulfate aerosol formation

The Benguela region has substantial uncertainties in DMS concentrations in surface seawater and the corresponding
emission fluxes owing to the limited availability of biogenic sulfur measurements. To investigate the sensitivity of
DMS emission fluxes to changes in surface seawater DMS concentrations, we conducted two simulations with DMS
concentrations from Lana et al. (2011) and Galí et al. (2018) (see Sect. 2.1). The standard results presented thus far
were conducted using the Lana dataset.
In the southern Benguela, south of approximately 27° S, marked upwelling during the austral summer (Shannon and
Nelson, 1996; Hutchings et al., 2009) promotes phytoplankton growth and elevates DMS emissions. Although the
Lana dataset indicates that DMS emission fluxes over the Sc sub-domain peak in January (Fig. A4 of the Appendix),
coinciding with this phenomenon, it lacks clear seasonality for the remaining months. In contrast, satellite-based DMS
estimates from Galí show pronounced emissions throughout the austral summer (JFND), as shown in Fig. A4. Both
datasets concur in magnitude for January and February, a period with better data coverage in the Lana et al. (2011)
climatological data set over the domain. However, the Lana dataset DMS emissions are up to 38% less during
December, while 51% higher in July relative to the Galí dataset (Ghahreman et al., 2019). This suggests the marine
contribution to sulfate in our standard simulation using the Lana dataset may be underestimated from October through
December (encompassing two months of the peak DMS season) and overestimated from March through August (Fig.
A4).
The ongoing discovery of complexities within DMS oxidation mechanisms, along with the incomplete incorporation
of these mechanisms into atmospheric chemistry models, further contributes to uncertainties in predicting the impact
of DMS emissions on aerosols and climate (Faloona, 2009; Quinn and Bates, 2011; Carslaw et al., 2013). Chen et al.
(2018)  highlighted the impacts of changes to DMS chemistry in the GEOS-Chem model, integrating a series of



345 multiphase sulfur oxidation mechanisms and two DMS intermediates, which led to a decrease in the global DMS

346 burden, thereby decreasing $SO_2$ and sulfate levels. On the other hand, Novak et al. (2021) found that the cloud uptake

347 of hydroperoxymethyl thioformate (HPMTF), a newly identified oxidation product of DMS (Wu et al., 2015; Veres

348 et al., 2020), lowers near-surface $SO_2$ concentration while elevating sulfate concentration in the model. Most recently,

349 Tashmim et al. (2024) implemented an advanced DMS oxidation mechanism in GEOS-Chem that incorporates the

350 latest developments in DMS chemistry, including those previously mentioned, which led to a lower $SO_2$ mixing ratio

351 (~70%) and a higher $SO_4$ mixing ratio (~35%) over the SEA during austral summer. Thus, an improved representation

352 of DMS emissions and oxidation chemistry in the model could enhance the sulfate aerosol estimations during the peak

353 DMS season. This refinement may address model underestimates of aerosol concentrations during this period (Sect.

354 3.1.1).

**355 3.3.2 Exploring the impact of marine organic aerosol emissions on organic aerosol concentrations**

356 Beyond marine sulfate and sea-salt aerosols, organic matter also makes a significant contribution to marine aerosol

357 mass (Middlebrook et al., 1998; Oppo et al., 1999; Russell et al., 2010). Notably, substantial concentrations of organic

358 carbon aerosols have been observed in marine regions, particularly during periods of intense biological activity

359 (O'Dowd et al., 2004). These aerosols can also increase CCN, affecting cloud properties and radiative balance (Arnold

360 et al., 2009; Gantt and Meskhidze, 2013). However, the standard GEOS-Chem model does not account for these

361 organic aerosol emissions. We analyzed the impact of marine POA on cloud-altitude aerosols over the SEA by

362 incorporating POA emissions based on satellite-derived chlorophyll-a concentrations (Gantt et al., 2015; See Sect.

363 2.1) in the model.

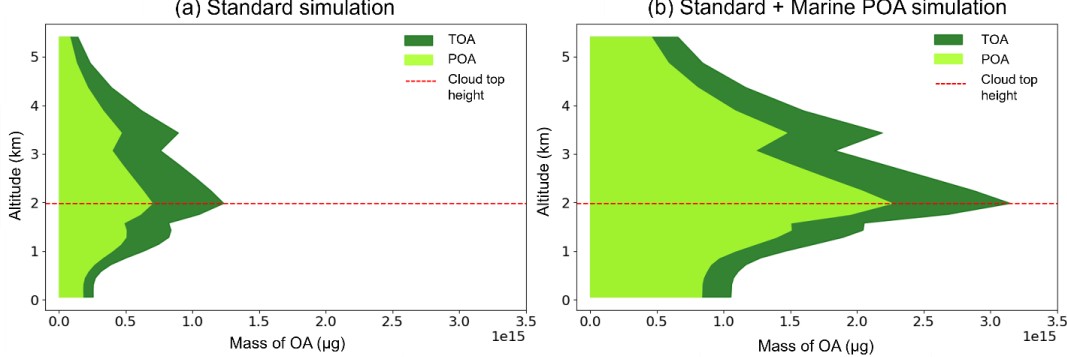

364

**Figure 9: Vertical distribution of organic aerosol mass during November 2017, the month of maximum discrepancy between the standard and MPOA simulations, over the Sc sub-domain (0–35° S, 20° E–20° W). Left: mass profile for total organic aerosols (TOA) and primary organic aerosols (POA) under standard simulation conditions (Std); right: when marine primary organic aerosol (MPOA) emissions are included (Std + MPOA). The red dashed line indicates the typical maximum cloud top height.**



We find that the inclusion of MPOA emissions consistently resulted in higher organic aerosol mass, with the greatest
increase in November. Figure 9 shows the vertical distribution of total organic aerosols (TOA) mass and POA mass
(including MPOA and other POA sources) with and without MPOA emissions during this month. Similar to our earlier
vertical profile analysis (refer to Sect. 3.2.2), we find that the maximum organic aerosol mass occurred at the highest
cloud top height (2 km). The Standard + MPOA simulated peak total organic aerosol mass was approximately three
times higher than that in the Standard simulation, highlighting the potential contribution of marine sources to total
organic aerosol mass concentrations. However, during the biomass burning season, the sensitivity simulation showed
only a minimal increase, indicating that it does not adequately address the model's underestimation (refer to Fig. 5).
Gantt et al. (2015) demonstrated that including MPOA emissions in GEOS-Chem reduced the normalized mean bias
(NMB) of surface organic aerosol concentrations at coastal sites by 67%. Additionally, Pai et al. (2020) noted that
without a marine POA source, the model fails to accurately reproduce lower-tropospheric concentrations over oceans,
although the marine POA scheme might be biased high. Despite the limitations of a chlorophyll-based
parameterization like the one used here in providing mechanistic understanding of the seasonal and geographical
variability of organic matter emissions from sea spray (Burrows et al., 2022), our findings suggest that MPOA may
play a role in aerosol-cloud interactions outside of the biomass burning season, in addition to marine-derived sulfate
from DMS (Sect. 3.2).
**3.3.3 Impacts of uncertainties in biomass burning emissions of SO$_2$**
To assess the impact of uncertainty in biomass burning emissions of SO$_2$ on the relative contribution of marine vs.
land sources to aerosol, we performed a sensitivity analysis using two alternative inventories, QFED and GFAS (see
Sect. 2.1 and Table-A2). The standard simulations, as detailed in Sect. 2.3, use the default biomass burning inventory
in GEOS-Chem, GFED. The GFAS inventory SO$_2$ and CO emissions over the domain are constant in time, aligning
with QFED during the non-biomass burning months (Fig. A5). We find that CO emissions from GFED and QFED
align closely; however, there is a notable difference in SO$_2$ emissions between the two inventories (Fig. A5). These
discrepancies likely originate from variations in SO$_2$ emission factors employed by each inventory. In July, which
exhibits the largest difference between the two inventories, peak SO$_2$ emissions in QFED are almost five-fold higher
than those in GFED. This discrepancy leads to a 25% increase in sulfate aerosol concentrations at cloud altitudes
relative to the standard results using GFED (not shown). Consequently, the contribution of marine sulfate to total
sulfate (see Sect. 3.2.1) may further decrease during the peak biomass burning season if QFED is used, highlighting
the sensitivity of aerosol source attributions to the selected biomass burning inventory.
**4 Implications**
In this study, monthly marine sulfate constitutes between 10.3% and 57.7% of total sulfate within the cloud height,
peaking during the high DMS emission period. However, the default Lana et al. (2011) climatology largely
underestimates DMS emissions during the austral summer (November and December) by up to 38%, compared to the
satellite-derived estimates from Galí et al. (2018). Moreover, improvement of DMS chemistry in the model by
incorporating new oxidation mechanisms and intermediate products could shift the balance towards increased sulfate



aerosol production (with Tashmim et al., 2024 suggesting an increase of up to 35% over the SEA).  Marine primary
organic aerosol emissions may also contribute substantially to the organic aerosol mass during the peak primary
production period (JNFD), highlighting the importance of marine contributions to overall aerosol concentrations.
Meanwhile, discrepancies in $SO_2$ emissions from biomass burning can increase sulfate aerosol from biomass burning
by up to 25%. These changes would improve the model underestimate of AOD relative to AERONET observations;
however, observations of aerosol composition outside of August-September are very limited and this is a large gap.
Our results suggest marine-sourced sulfate and organics significantly influence aerosol loading and composition in
the SEA, particularly during the non-biomass burning period. Accurately characterizing the seasonal dynamics of
aerosols within cloud heights is imperative for quantifying aerosol-cloud interactions and understanding the dynamics
of marine aerosols in the SEA region, where uncertainties in aerosol radiative forcing are most pronounced. This
understanding is essential for improving the reliability of climate models in areas critical to both regional and global
climate dynamics.
**5 Conclusion**
Aerosols over the southeast Atlantic strongly influence global climate dynamics due to the presence of persistent
stratocumulus clouds and large uncertainties in aerosol-cloud interactions. However, precisely representing these
interactions in global climate models remains challenging, in part due to sparse available observations, especially
outside of the biomass burning season. In this study, we employed the GEOS-Chem chemical transport model to
assess the aerosol composition at cloud-relevant altitudes (0–2 km) and identify the sensitivities to marine emissions
and chemistry in the southeast Atlantic. This analysis aims to enhance our understanding of the role of marine aerosols
and the associated uncertainties affecting aerosol-cloud interactions within this climate-sensitive region.
We performed nested grid simulations with a 0.5° x 0.625° horizontal resolution and evaluated the model against
ground-based and aircraft campaign observations throughout 2017. We analyzed results for three seasonal periods
with distinct dominant processes including (a) the high DMS emission season (JFND), (b) the peak biomass burning
season (JASO), and (c) the transitional season (MAMJ). Our analysis showed that simulated monthly average aerosol
optical depth (AOD) exhibits the strongest correlation (R = 0.92) with the AERONET AOD observations during the
JASO season. However, the model generally underestimates AOD throughout the year, except in the JFND period.
These underestimations are primarily due to limitations in representing natural aerosol emissions, transatlantic aerosol
transport, particle mixing states, and the oxidation levels of organic aerosols. Moreover, a comparison of aerosol
speciation measured at Ascension Island during the LASIC campaign indicates that the model consistently
underestimates sulfate aerosols. We further evaluated the simulated vertical profile of aerosol mass concentrations
and composition against measurements from the ORACLES and CLARIFY campaigns. These comparisons showed
that sulfate aerosols were underestimated by 19% at cloud-relevant altitudes of 0–2 km by both campaigns. However,
discrepancies increase with altitude, reflecting challenges in accurately modeling high-altitude aerosol concentrations.
Analysis of seasonal mean aerosol composition at cloud height showed that organic aerosols predominate during
JASO (63%) and MAMJ (51%), while sulfate aerosols are most prevalent (41%) during the austral summer (JFND).



Given the prominence of sulfate as a marine sourced aerosol in remote oceanic environments, we investigated the
processes influencing the sulfate aerosol concentrations in our domain. Throughout the year, anthropogenic sources
and oceanic DMS emissions are the primary atmospheric sulfur contributors. Spatial mapping across the sub-domain
(0–35° S, 20° E–20° W) showed high sulfate concentrations (up to 3µg m$^{-3}$) at cloud height during the peak biomass
burning season (JASO), primarily from savannah fires in southern Africa. Despite this, sulfate aerosols only account
for 18% of the total aerosol mass in JASO.
Sulfate, primarily from marine sources, is the dominant aerosol at cloud-relevant altitudes during JFND in the model
(up to 69% marine contribution); however, significant uncertainties regarding the treatment of DMS persist that may
affect this finding. To assess the impact of these uncertainties on sulfate aerosols, we compared DMS emission fluxes
from Lana et al. (2011) climatological data and Galí et al. (2018) satellite-based estimates of surface seawater DMS
concentrations. The limited spatial and temporal coverage of the Lana dataset across our domain resulted in a 51%
overestimate in emissions in July and a 38% underestimate in December relative to Galí. Moreover, improvement of
DMS chemistry in the model by incorporating new oxidation mechanisms and intermediate products could shift the
balance towards increased sulfate aerosol production (with Tashmim et al., 2024 suggesting an increase of up to 35%
over the SEA). Additionally, emissions of marine primary organic aerosols during the peak primary production period
(JNFD) may substantially contribute to the mass of organic aerosols which can also act as CCN. This emphasizes the
critical role of marine sources in influencing aerosol concentrations, even in oceanic regions impacted by large
seasonal biomass burning. Variations in SO$_2$ emissions from biomass burning could potentially increase sulfate
aerosol concentrations at cloud altitudes by up to 25%. Addressing these discrepancies is essential for improving the
model's underestimation of AOD and aerosol concentrations compared to observations.
This study highlights the importance of constraining marine emissions and their chemical transformations by
incorporating satellite-retrieved datasets and extending field campaign efforts during non-biomass burning periods.
Such initiatives are essential to accurately characterize seasonal aerosol dynamics at cloud heights and to improve our
understanding of aerosol-cloud interactions in regions with persistent low-altitude clouds. These advancements could
substantially minimize uncertainties in model estimates of radiative forcing and enhance the reliability of climate
model projections in the southeast Atlantic region.










**Appendix A**
**Table A1:** Configuration of sensitivity analysis simulations

| Simulations | Resolution |
|---|---|
| Marine sulfur emissions only | 0.5° x 0.625° |
| DMS emissions | 4° x 5° |
| Biomass burning inventories | 4° x 5° |
| Marine primary organics | 0.5° x 0.625° |



**Table A2:** AERONET site information and the average value (±1 standard deviation) for $AOD_{550}$ per site are shown.

| Site | Latitude (°) | Longitude (°) | Months of data availability for 2017 | Daily Average $AOD_{550nm} \pm 1$ SD |
|---|---|---|---|---|
| Ascension Island | -7.976 | -14.415 | 7 | 0.18 ± 0.04 |
| Gobabeb | -23.562 | 15.041 | 12 | 0.10 ± 0.04 |
| HESS | -23.273 | 16.503 | 10 | 0.08 ± 0.04 |
| Henties_Bay | -22.095 | 14.26 | 3 | 0.25 ± 0.02 |
| Lubango | -14.958 | 13.445 | 9 | 0.16 ± 0.05 |
| Namibe | -15.159 | 12.178 | 8 | 0.33 ± 0.12 |
| Simonstown_IMT | -34.193 | 18.446 | 7 | 0.05 ± 0.03 |
| Upington | -28.379 | 21.156 | 8 | 0.08 ± 0.06 |
| Windport | -19.366 | 15.483 | 10 | 0.15 ± 0.08 |








**Table A3:** Seasonal variation of percentage of monthly mean percent contribution of marine sulfate within cloud
height

| Month | Percentage of marine sulfate |
|---|---|
| January | 57.7 |
| February | 54.8 |
| March | 25.3 |
| April | 26.6 |
| May | 15.3 |
| June | 15.0 |
| July | 14.8 |
| August | 14.7 |
| September | 10.3 |
| October | 22.4 |
| November | 39.1 |
| December | 44.3 |








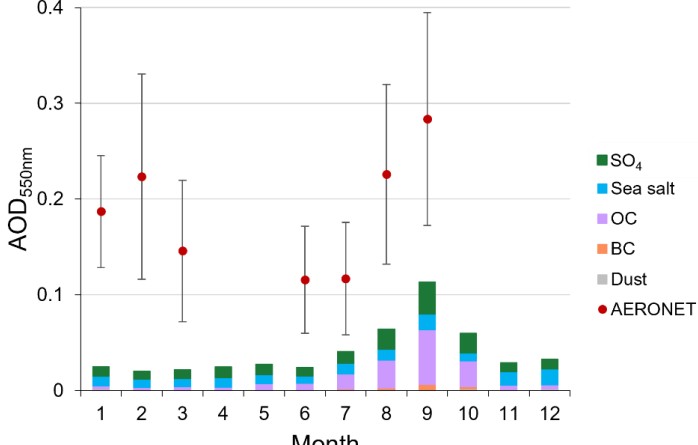


**Figure A1:** Comparative analysis of aerosol optical depth at 550 nm ($AOD_{550nm}$) for Ascension Island in 2017. The red dots present the measured mean monthly AOD values, with vertical error bars illustrating the range of $AOD_{550nm}$ measurements captured by the AERONET ground station. The stacked bars represent the GEOS-Chem model's simulated AOD values, with each layer corresponding to the major aerosol components, such as sulfate ($SO_4$), sea salt, organic carbon (OC), black carbon (BC), and dust, providing insight into the model's aerosol composition representation.



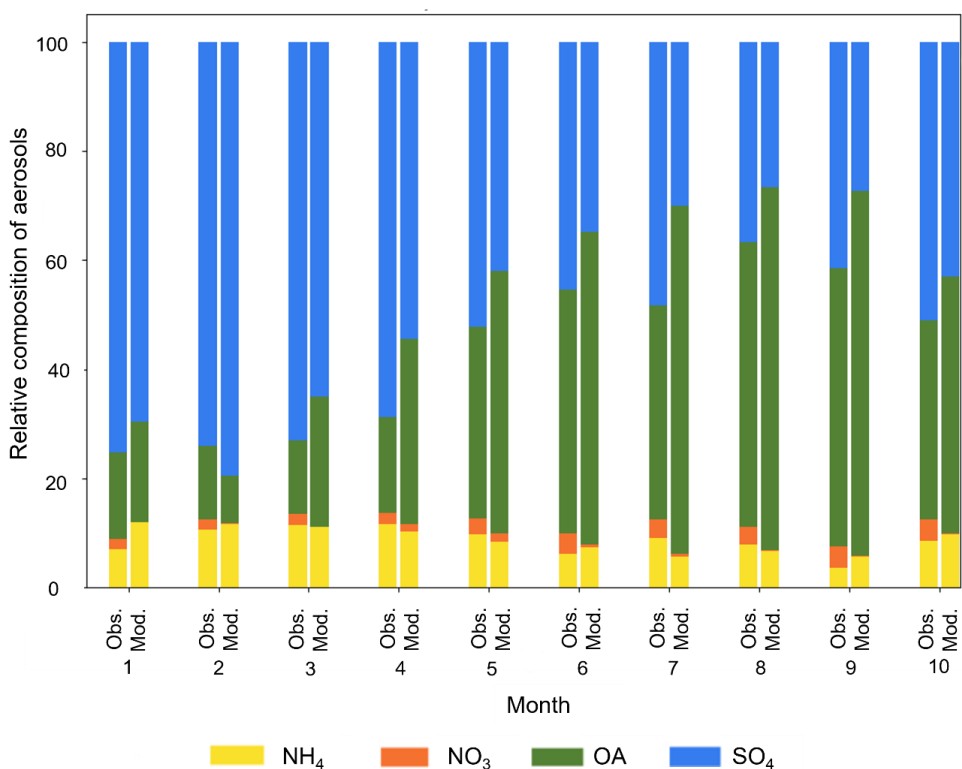

**Figure A2:** Comparative analysis of the relative aerosol composition at Ascension Island in 2017. The stacked bars on the left depict observations of chemical composition taken during the LASIC campaign at the ARM facility on Ascension Island, utilizing an aerosol chemical speciation monitor (ACSM) at 341 meters. The bars on the right illustrate the GEOS-Chem simulated aerosol composition at the same altitude. Each segment of the stack represents different aerosol components: ammonium ($NH_4$), nitrates (NIT), organic aerosols (OA), and sulfate ($SO_4$).





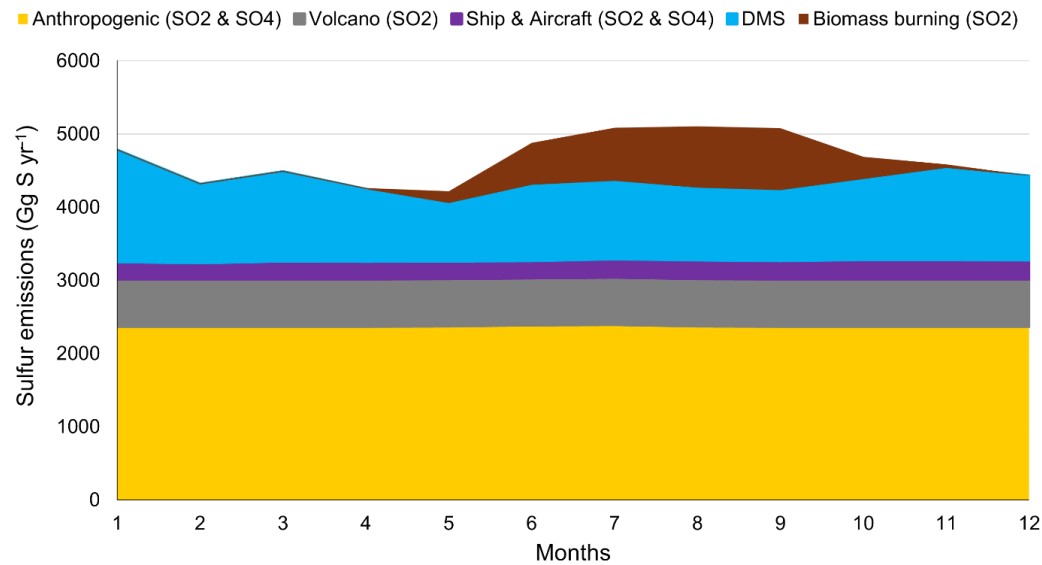

501

**Figure A3:** Stacked area chart of monthly total sulfur emissions by source for 2017 across the study domain (0–40°
S, 40° E–20° W) in gigagrams of sulfur per year (Gg S yr⁻¹). Sources are indicated by color and encompass
anthropogenic activities, volcanic activity, ship and aircraft emissions, biomass burning and natural emissions of
dimethyl sulfide (DMS).







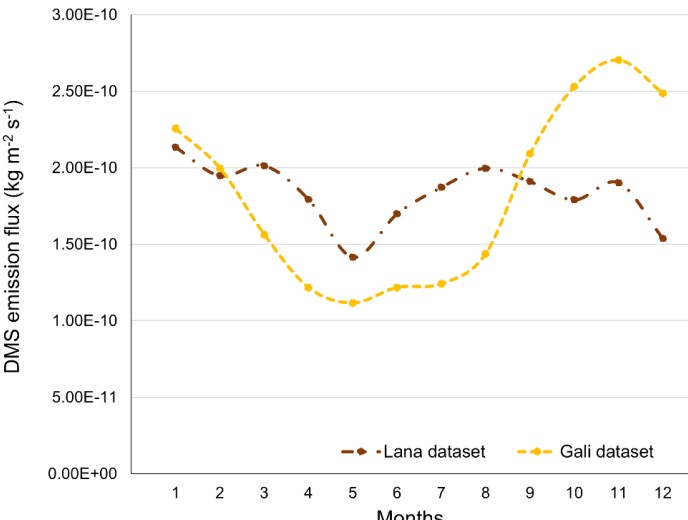


**Figure A4:** Monthly DMS emissions over the stratocumulus sub-domain (0–35° S, 20° E–20° W) using two distinct datasets for surface seawater DMS concentrations. The brown dashed line presents emissions calculated using Lana et al. (2011) climatology, which compiles data across 1972-2009 from multiple sources. In contrast, the yellow dashed line depicts emissions based on satellite-derived estimates of surface seawater DMS concentrations (Galí et al., 2018).

515

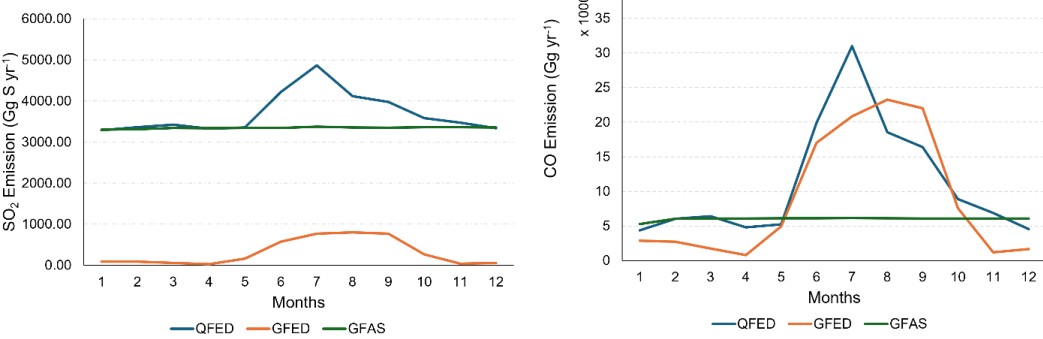

516

**Figure A5:** Comparison of biomass burning emissions across various inventories, namely GFED, QFED, and GFAS across the domain (0–40° S, 40° E–20° W). The panels depict the interannual variability of biomass burning emissions, with the left panel illustrating sulfur dioxide ($SO_2$) emissions, and the right panel displaying carbon monoxide (CO) emissions. Both GFED and QFED indicate similar emission trends; however, GFED exhibits lower $SO_2$ emission magnitudes compared to QFED. GFAS presents emission magnitudes similar to QFED during non-biomass burning period.



**Author contributions**

MH, HMH, and RMG designed the research. MH conducted the model simulations, analysis, and visualization, with expert advice from HMH. MH drafted the manuscript, which was then revised by all co-authors.

**Competing interests**

At least one of the (co-)authors is a member of the editorial board of Atmospheric Chemistry and Physics.

**Acknowledgements**

Mashiat Hossain and Hannah M. Horowitz gratefully acknowledge Michael Diamond for discussions on cloud-relevant altitudes over the southeast Atlantic region.

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
