# Peer review of "Quantifying the Impacts of Marine Aerosols over the Southeast"

_EGUsphere, 2024_

## Referee Comment (RC2)

Review of "Quantifying the Impacts of Marine Aerosols over the Southeast Atlantic Ocean using a chemical transport model: Implications for aerosol-cloud interactions"

This paper uses the GEOS-Chem model and surface and aerial campaign measurements to investigate the contributions and sources of sulfate and organics to marine aerosol in boundary layer and free-tropospheric aerosol in the Southeast Atlantic. Based on model comparisons to the in situ measurements, it was found that GEOS-Chem underestimates sulfate aerosol due to poorly represented sulfate/DMS fluxes, which can have important implications for model CCN and their interactions with clouds. Large differences in model and surface-retrieved AOD were also attributed to poorly represented natural aerosol emissions, biomass burning transport, and the assumed mixing state of aerosol. This work is very interesting and quite thorough in its analyses and presentation of results. It provides important context and guidance to improve aerosol representation in models. I feel that the paper is suitable for ACP and should be published after considering the following minor comments and suggestions.

**Comments:**

- Lines 48-50: there is also a sulfate particle flux, in addition to salts and organic matter, that is attributable to the production primary marine aerosols. See Russell et al. (2023).

- This curious review wonders how the ship sulfate was prescribed in the model? (lines 104)?

- Table A1: Can the authors provide a bit more context to this table? I know the long details are provided in the main text, but for quick reference it could useful to have a brief note on things like: surface climatology or satellite-derived (for DMS), the different constraints on BB inventories, etc...and the references for these.

- Table A2: How do the monthly averages of AOD at each site compare for the "background" conditions (Nov-Jul) and the austral spring biomass burning affected periods (Aug-Oct) and are the differences statistically significant? If so, can this be added to the table? I think Fig. 4 may show this, but there are a lot of points and scatter.

- LASIC ACSM data: did the authors use the composition-dependent collection efficiency (CDCE) mass concentrations in this work? If so, please specify. The aerosol measured are also non-refractory, so please specify that as well.

- Figure 2: It may help the reader to add a symbol indicating Ascension Island on this map and in the legend.

- Figure 4: how large are the standard deviations for each data point shown here (taken from monthly mean values across all sites)?

- Fig. A1: Are the modeled AOD averaged for grid points around Ascension Island or do they represent the entire domain? Please specify in the caption. If the entire domain was used, are you also able to show grid point values for near/around Ascension Island as this was the site used for comparison?

- Line 202-203: Observational evidence of predominantly internally-mixed and aged biomass burning particles in the Southeast Atlantic from aircraft measurements has been shown by Dang et al. (2022), please include.

- Fig. A2 caption: nitrates in the figure legend are identified as "NO3" while in the caption nitrates are "NIT". Please correct for consistency.

- Fig. A2: conventional ACSM/AMS composition coloring is typically, organics (green), nitrate (blue), sulfate (red), and ammonium (orange). For consistency with this convention and to not confuse the coloring scheme in Fig. A1, it is preferred that the authors stick with the ACSM/AMS convention.

- Figure 6: similar to Fig. A2 caption, please correct NIT and NO3 for consistency.

- Line 325-326: Has previous work described or quantified the "substantial uncertainties in DMS concentrations…"? I feel as though a citation is needed here.

**Minor edits:**

- Line 47: "…, leading to [the] largest uncertainty of aerosol radiative forcing…"

- Line 200: delete, "during" at the end of the sentence.

References

Dang, C., Segal-Rozenhaimer, M., Che, H., Zhang, L., Formenti, P., Taylor, J., Dobracki, A., Purdue, S., Wong, P., Nenes, A., Iii, A., Coe, H., Redemann, J., Zuidema, P., Howell, S., and Haywood, J.: Biomass burning and marine aerosol processing over the southeast Atlantic Ocean: a TEM single-particle analysis, Atmospheric Chemistry and Physics, 22, 9389-9412, 10.5194/acp-22-9389-2022, 2022.

Russell, L., Moore, R., Burrows, S., and Quinn, P.: Ocean flux of salt, sulfate, and organic components to atmospheric aerosol, Earth-Science Reviews, 239, 10.1016/j.earscirev.2023.104364, 2023.

---

## Author Comment (AC1)

**Response to reviewers – "Quantifying the Impacts of Marine Aerosols over the Southeast Atlantic Ocean using a chemical transport model: Implications for aerosol-cloud interactions"**

Thank you to both reviewers for their insightful and constructive comments. Your suggestions have contributed to improving the clarity and quality of the manuscript. We provide our responses below in blue text, with bold blue text for revised sentences from the manuscript. Line numbers in the responses correspond to the track changes version of the revised manuscript.

**Comments from Reviewer #1**

Overall, the paper and analysis are solid. As noted in the Conclusions, "This study highlights the importance of constraining marine emissions and their chemical transformations by incorporating satellite-retrieved datasets and extending field campaign efforts during non-biomass burning periods." This is a useful contribution to the literature on the aerosols that influence clouds in the SE Atlantic stratocumulus region – where most of the focus has been on the influence of biomass burning emissions. The inclusion of the section on Uncertainties, and quantifying how different emission inventories and marine organic emission affect the papers' conclusions, is very good.

We appreciate the reviewer's positive feedback on our work.

I have just a few more substantial comments, and then a set of smaller points. My recommendation is that the paper be published after minor revision, per the following comments:

More substantial points:

While there's no harm in comparing the modeled and observed AODs, the stated focus of the paper is on the role of marine aerosols in aerosol-cloud interactions over the SE Atlantic. In this region, especially during the season when in-situ observations are available for testing the model, the majority of AOD is above-cloud. The real information informing model biases of boundary-layer / in-cloud aerosols comes from the vertically-resolved observations from CLARIFY and ORACLES. I think Section 3.1.1 could be significantly shortened, to focus on just giving the big picture of sources of aerosols over the region.

Thank you for your comment. We revise Section 3.1.1 to focus more on the main aim of the paper — marine aerosol impacts within boundary layer. Specifically, we revise the following sentence (lines 211-212):
**"Excluding these two sites, improves the model's correlation to 0.67 (p = 0.55) and reduces the NMB to 4.7%."**

Additionally, we omit the following sentence: "Furthermore, Table 2 shows that the model underestimates AOD during JASO by 18.6% (NMB) across the domain."

Lastly, we revise and move the following sentences to Section 3.1.2:
**"This underestimation may also be affected by the model's bulk aerosol scheme, which inadequately captures the optical properties of aerosols and is compounded by a low relative humidity bias (Zhai et al., 2021). The bulk scheme also assumes all aerosols are externally mixed, which contrasts with the variable degree of particle mixing states in the atmosphere (Yu et al., 2012; Dang et al., 2022). Moreover, studies like Hodzic et al. (2020) using NASA ATom aircraft data indicate that GEOS-Chem substantially underestimates oxidation levels of organic aerosols in remote areas, which could affect estimates of their burden and optical properties."**

In contrast, given the focus of the paper it seems to me that Figure A3, showing the seasonal emissions sources for sulfur, and Table A3, giving the fractional contribution of marine sulfate at cloud height by month, should be shown in the main part of the paper. Also possibly Figure A4. I think the AOD plots could instead be moved to the Appendix if wanting to limit the number of figures in the main paper.

We move Figures A3 and A4, as well as Table A3, to the main paper and rename them accordingly.

Regarding Figure A3: The Anthropogenic, Volcanic and Ship and Aircraft sources of sulfate appear to have no seasonality. Particularly for the Anthropogenic contribution – which is the largest of all emissions sources – this seems odd at best. Is this due to a lack of better information? An assumption? If this is based on not knowing the seasonal variability, versus there not being any actual seasonal variability, this needs to be at least acknowledged, as it for sure will affect conclusions about the relative contributions of each in different seasons.

We add information on apparent lack of seasonality within various inventories representing the various sources of sulfate in the model. This limitation has been acknowledged in the first paragraph of Section 3.2.1 (modified text underlined):
**"However, the model default CEDS inventory (Hoesly et al., 2018) fails to capture the seasonality of these emissions due to absence of regional inventories and reliance on the global datasets such as the International Energy Agency (IEA) energy statistics. The anthropogenic emissions are followed by DMS emissions from the ocean, which become more pronounced during the austral summer, peaking in January. Additionally, biomass burning contributes to $SO_2$ emissions seasonally, becoming the 3rd most important source of total sulfur emissions during July - September (Fig. A3). In contrast, sulfur contributions from volcanic, shipping, and aircraft emissions remain minimal and constant year-round, reflecting assumptions of static fuel burned and emission levels across inventories."**

Lines 200-206: Here, it is posited that several known biases in GEOS-Chem could be responsible for the 26.5% NMB in AOD during the biomass burning season. However, in Figure 5 it is pretty clear that there are significant low biases in GEOS-Chem in a) the amount of aerosol at higher altitudes, and significant low biases in the amount of NO3 – in addition to low biases in OA. These are likely bigger factors than inaccuracy in the aerosol optical properties of the aerosol. It would make more sense to discuss the source of model biases after discussing Figure 5, because then you can put known model issues in the context of the observed biases.

As mentioned in response to the first comment, the discussion of AOD biases in the model has been rephrased and moved to Section 3.1.2, following the discussion of Figure 5 (lines 238-245):
**"These significant low biases in aerosol concentrations at higher altitude likely contribute to the model's underestimation of AOD during the biomass burning period (see Section 3.1.1). This underestimation may also be affected by the model's bulk aerosol scheme, which inadequately captures the optical properties of aerosols and is compounded by a low relative humidity bias (Zhai et al., 2021). The bulk scheme also assumes all aerosols are externally mixed, which contrasts with the variable degree of particle mixing states in the atmosphere (Yu et al., 2012; Dang et al., 2022). Moreover, studies like Hodzic et al. (2020) using NASA ATom aircraft data indicate that GEOS-Chem substantially underestimates oxidation levels of organic aerosols in remote areas, which could affect estimates of their burden and optical properties."**

Line 225: a low bias of 20->55% is not "moderate agreement"… This is a significant low bias.

We revise the sentence as follows (lines 235-236):
**"At mid-altitudes (2–4 km), the model is biased low, with NMB values spanning -19% (ORACLES) to -57% (CLARIFY)."**

Figures 3 and Figure 7: Please add latitude and longitude to these maps. The longitude markers in Figure 3 need fixing as they are given as negative in both the E and W directions. Also, these maps don't cover the 0-30S, 20E-20W box specified in the text & Fig 6 caption – so they either should, or that box should be indicated in the figure.

We have updated Figure 3 to display the entire domain, along with the sub-domain (0–35° S, 20° E–20° W), which is highlighted by a green dashed box in Fig. 3(a). Thank you for pointing out the mistake in the longitude markers in Figure 3; this has also been corrected. Latitude and longitude markers have also been added to Figure 7.

On lines 86-87, it says that in the model: "Organic aerosol follows the "simple" scheme which treats primary organic aerosol (POA) as non-volatile and includes irreversible direct yield of SOA from precursors." i.e.: The model represents the contribution of both POA and SOA to total OA in the model. However lines 356-357 then lines 360-361, say: "Beyond marine sulfate and sea-salt aerosols, organic matter also makes a significant contribution to marine aerosol … However, the standard GEOS-Chem model does not account for these organic aerosol emissions. We analyzed the impact of marine POA on cloud-altitude aerosols over the SEA by incorporating POA emissions based on satellite-derived chlorophyll-a concentrations."  Can you please better explain what is not being accounted for in the model, that this analysis is looking at? Does the model account for land-based emissions that contribute to POA and SOA but not ocean-based? I was confused by this.

The model accounts for land-based POA and SOA, but marine POA emissions are not included in the standard simulations. Section 3.3.2 explores the impact of adding marine-based primary organic aerosol emissions on the total organic aerosol concentrations. We have made corrections (line 394-397) to clarify this as follows:
**"However, the standard GEOS-Chem model does not account for these marine organic aerosol emissions. Here, we analyzed the impact of marine POA on cloud-altitude aerosols over the SEA by incorporating marine POA emissions…"**

Lines 431-432:  It's asserted that "These underestimations are primarily due to limitations in representing natural aerosol emissions, transatlantic aerosol transport, particle mixing states, and the oxidation levels of organic aerosols." However, these are really hypothesized sources of bias; the analysis in the paper doesn't involve analyzing the contributions of each of these to biases in simulated AOD, so I don't think it should be asserted here that this is the case, as no evidence given to support this. Again, I think the part of the analysis that focuses on AOD really doesn't add much to addressing the main question of the paper, which is how marine sources contribute to sulfate and organics at cloud altitudes.

We revise the paragraph to place the discussion on AOD underestimation towards the end, emphasizing that aerosol underestimation is a significant factor contributing to this issue, as supported by the evidence provided. The revised text now reads as follows (lines 469-472):
**"The underestimate of sulfate aerosols at lower altitudes (0-2 km), coupled with an underestimate of other aerosol at higher altitudes (4-8 km), likely contributes to the overall low bias in modeled AOD. The misrepresentation of simulated natural aerosol emissions and transatlantic aerosol transport may be responsible for these underestimates."**

**Smaller points:**

Abstract, pg 1, lines 17-18: "At these altitudes, organic aerosols (63%) dominate during the biomass burning period, while sulfate (41%) prevails during austral summer, when dimethylsulfide (DMS) emissions peak in the model." It's not clear whether these numbers are also coming from the model. If all are from the model, reword to e.g.: "At these altitudes, in the model organic aerosols (63%) dominate during the biomass burning period, while sulfate (41%) prevails during austral summer, when dimethylsulfide (DMS) emissions peak." If not from the model, where are these numbers from?

Thank you for the suggestion. The numbers mentioned do indeed come from the model. We have implemented the suggested rewording in the abstract (lines 17-18) to clarify this.

Abstract, pg 1, lines 19-21: "Sensitivity analyses indicate that refining DMS emissions and oxidation chemistry may increase sulfate aerosol produced from marine sources, highlighting their overall importance." It's not clear to me how demonstrating that changing the oxidation chemistry has an effect on sulfate production indicates that DMS itself is necessarily important (important in what regard?). I'd say instead that this analysis highlights that there remains large uncertainty in the role of DMS emissions in marine boundary layer sulfate, which is important given that DMS appears to make a significant contribution to sulfate concentration in the Sc clouds in the SE Atlantic region.

We rephrase the sentence (line 19-22) as follows:
**"Sensitivity analyses indicate that refining DMS emissions and oxidation chemistry may increase sulfate aerosol produced from marine sources, highlighting that there remains large uncertainty in the role of DMS emissions in the marine boundary layer."**

Abstract, pg 1, lines 22-24: "This study underscores the imperative need to refine marine emissions and their chemical transformations to better predict aerosol-cloud interactions and reduce uncertainties in aerosol radiative forcing over the southeast Atlantic." Nothing given up to this point in the abstract (or in the paper) quantifies how biases in the contribution of marine sulfate actually affect aerosol-cloud interactions – only that marine aerosol account for a large fraction of the sulfate at cloud altitudes outside of the biomass burning season. So I don't think it can be asserted that this is a significant source of bias or uncertainty in aerosol-cloud interactions over the SE Atlantic. It might indeed be; but the analysis here does not allow one to conclude that.

We clarify our statement by rephrasing it (lines 23-25):
**"This study underscores the imperative need to refine marine emissions and their chemical transformations, as aerosols from marine sources are a major component of total aerosol at cloud-relevant altitudes and may impact uncertainties in aerosol radiative forcing over the southeast Atlantic."**

Pg 1, lines 30-31: "However, aerosol radiative forcing in the region exhibits highest uncertainty and one of the largest intermodel spread," Across what set of models? And relative to the aerosol radiative forcing in other stratocumulus regions? Or in clouds anywhere? This needs to be contextualized.

We rewrite the sentence to add contextual relevance (lines 31-35):
**"However, this region exhibits highest uncertainty in aerosol radiative forcings in the AeroCom intercomparison across CMIP5 general circulation models (GCMs) and chemical transport models (Stier et al., 2013). This uncertainty is primarily driven by challenges in accurately representing cloud fraction, aerosol-cloud properties, and vertical structure, both in the presence and absence of smoke (Stier et al., 2013; Doherty et al., 2022)."**

Pg 2, lines 36-37: "and sources of uncertainty affecting aerosol composition" The paper focuses on this for the boundary layer/marine aerosol only – not for the biomass burning aerosol aloft, which is important since the elevated biomass burning aerosol layer can significantly contribute to aerosol-cloud interactions for the year.

To clarify, we revise the sentence as follows (lines 35-37; modifications are underlined):
**"In this study, we investigate the role of marine aerosols and sources of uncertainty affecting aerosol composition within the boundary layer, particularly in this critical region of aerosol-cloud interactions over the SEA."**

Pg 2, lines 47-48: "leading to largest uncertainty of aerosol radiative forcing within climate models" As above, this needs context. Are you really saying that the formation of sulfate from DMS is the largest source of uncertainty in aerosol radiative forcing? Such a strong statement would need to be supported by a more

recent set of publications than the 2013 reference cited, as climate models have evolved significantly in the past decade.

We revise the paragraph to provide clearer context with appropriate citations (lines 45-50):
**"Although DMS is a critical source of natural aerosols, contributing over 50% of natural gas-phase sulfur emissions (Chin et al., 1996; Kilgour et al., 2021), the exact mechanisms of DMS oxidation and subsequent formation of sulfate and MSA aerosol remain inadequately understood (Ravishankara et al., 1997; Barnes et al., 2006; Hoffmann et al., 2016). This gap in understanding contributes to substantial uncertainty in aerosol radiative forcing, which is highly sensitive to uncertainties in natural aerosols (Carslaw et al., 2013; Fung et al., 2022)."**

Lines 59-60 & 68-70: A very small point, but it would be helpful to the reader if the references given on lines 59-60 were mapped to the field campaigns given on lines 68-70. This could be done by giving (again) the citation for the overview paper of each campaign, e.g.: "(CLARIFY; Haywood et al., 2021)"

We add the citation as per recommendation.

Line 81: "as 10 minutes" --> "of 10 minutes"

Thank you for pointing it out – we correct it in the manuscript.

Lines 85 & 86: I am not sure what you mean by "follows". In terms of emissions? In the optical / size / other properties of these constituents?

Our aim was to indicate that the respective aerosols utilized the methodologies outlined in the corresponding literature for formation and partitioning within the model. However, we now rephrase the sentence (changes are underlined):
**"Organic aerosol is simulated using the "simple" scheme which treats primary organic aerosol (POA) as non-volatile and includes irreversible direct yield of SOA from precursors (Pai et al., 2020). The BC simulation follows the methodologies of Park et al. (2003) and Wang et al. (2014)."**

Lines 98-99: I assume that sea salt emissions are also wind-dependent!

Thank you for catching this. It has been corrected in line 100-101:
**"Sea-salt aerosol (SSA) emissions from the open ocean are both windspeed- (Gong et al., 2003) and sea surface temperature-dependent (Jaeglé et al., 2011)."**

Figure 3: There is clearly a peak in emissions / AOD in the vicinity of Johannesburg. The text around lines 164-168 comments on the other features of the AOD but not this one; it would be good to do so just briefly. (It's referenced when discussing Fig 3b, but not the other panels – and it's present in all three)

We add a sentence at the end of the paragraph to acknowledge the presence and source of the hotspot in the Gauteng province, along with appropriate references (lines 187-189):
**"Additionally, a year-round AOD hotspot is observed in northeastern South Africa (Gauteng province; Fig. 3), which is associated with elevated aerosol concentrations due to industrial and mining activities, as well as domestic fuel burning (Arowosegbe et al. 2021; Zhang et al. 2021)."**

Lines 173-174: "This increase, combined with dust emissions from the Namib desert, contributes to an AOD hotspot as depicted in Fig. 3a on the southwestern coast." Figure 7 shows the contribution from sulfate. Presumably you can therefore make a statement here about the relative contributions of sulfate vs dust to this 'hot spot'. Correct?

We modify the following sentence in Section 3.1.1 to clarify the relative contributions of sulfate and dust to the hotspot discussed (lines 181-183).

**"This increased sulfate (~20%), combined with dust (59%) emissions from the Namib desert, contributes to an AOD hotspot as depicted in Fig. 3a on the southwestern coast."**

Line 190: Not just a low correlation coefficient – but a negative correlation coefficient - !

We modify the sentence as (lines 208-210):
**"Negligible negative correlation coefficient (R = -0.058) with a positive bias (29.8%) is seen during the summer period (JFND), predominantly due to anomalies at two sites."**

Lines 199-200: There is an incomplete sentence here.

Thank you for noticing – this sentence has been omitted.

Lines 450-451: "The limited spatial and temporal coverage of the Lana dataset across our domain resulted in a 51% overestimate in emissions in July and a 38% underestimate in December relative to Galí." Unless you know that the Galí dataset is "truth", this would be better phrased as a difference between the two rather than a bias in the Lana dataset.

We rephrase the sentence in the text (lines 487-489):
**"We find that, within our domain, the Lana dataset emissions estimates are 51% higher in July and 38% lower in December compared to the Galí dataset."**

**Comments from Reviewer #2**

Review of "Quantifying the Impacts of Marine Aerosols over the Southeast Atlantic Ocean using a chemical transport model: Implications for aerosol-cloud interactions"
This paper uses the GEOS-Chem model and surface and aerial campaign measurements to investigate the contributions and sources of sulfate and organics to marine aerosol in boundary layer and free-tropospheric aerosol in the Southeast Atlantic. Based on model comparisons to the in situ measurements, it was found that GEOS-Chem underestimates sulfate aerosol due to poorly represented sulfate/DMS fluxes, which can have important implications for model CCN and their interactions with clouds. Large differences in model and surface-retrieved AOD were also attributed to poorly represented natural aerosol emissions, biomass burning transport, and the assumed mixing state of aerosol. This work is very interesting and quite thorough in its analyses and presentation of results. It provides important context and guidance to improve aerosol representation in models. I feel that the paper is suitable for ACP and should be published after considering the following minor comments and suggestions.

We thank the reviewer for their support for this paper.

Lines 48-50: there is also a sulfate particle flux, in addition to salts and organic matter, that is attributable to the production primary marine aerosols. See Russell et al. (2023).

Thank you for mentioning. We revise the sentence as follows (lines 50-53):
**"Additionally, marine aerosols comprise primary aerosols such as sea spray aerosols, which consist of salts, sulfate, and organic matter, released into the atmosphere primarily by the bubble-bursting process (O'Dowd and De Leeuw, 2007; Russell et al., 2010; Prather et al., 2013; Brooks and Thornton, 2018; Russell et al., 2023)."**

This curious review wonders how the ship sulfate was prescribed in the model? (lines 104)?

The ship-related sulfate emissions in the model are derived from the CEDS inventory, where fuel consumption estimates for shipping are based on a composite time series from various sources covering the period 1850-2012, including recent bottom-up estimates (Fletcher 1997, Smith et al. 2011, IMO 2014).

Since the national-level fuel data have little geographical relevance (e.g., the fuel is consumed over some international route), international shipping emissions are summed and reported only at the global level.

Table A1: Can the authors provide a bit more context to this table? I know the long details are provided in the main text, but for quick reference it could useful to have a brief note on things like: surface climatology or satellite-derived (for DMS), the different constraints on BB inventories, etc…and the references for these.

Table A1 has been revised, and now includes descriptions of the emission measurement mechanisms and the associated emission inventories, along with their references.

Table A2: How do the monthly averages of AOD at each site compare for the "background" conditions (Nov-Jul) and the austral spring biomass burning affected periods (Aug-Oct) and are the differences statistically significant? If so, can this be added to the table? I think Fig. 4 may show this, but there are a lot of points and scatter.

To provide clarification, we revise Table A2 to include the monthly AOD averages and standard deviations for individual sites across three distinct seasons discussed in the manuscript, namely: peak DMS emission period (JFND), peak biomass burning months (JASO), and transition period (MAMJ). A comparison of the monthly AOD averages between the background conditions (November to June) and the biomass burning period (July to October) indicates a statistically significant difference between the means. These findings have been updated in Section 3.1.1 (lines 202-205) as follows:
**"The comparison of monthly mean AOD across individual sites (see Table A2 in the Appendix) shows that, with the exception of Ascension Island, Gobabeb, and Upington, the mean AOD at the remaining sites during the biomass burning season (JASO) is atleast one standard deviation higher than the mean AOD in other seasons (JFND & MAMJ)."**

LASIC ACSM data: did the authors use the composition-dependent collection efficiency (CDCE) mass concentrations in this work? If so, please specify. The aerosol measured are also non-refractory, so please specify that as well.

The initial comparison did not take into account the composition-dependent collection efficiency (CDCE) mass concentrations. Figure A2 has now been updated with aerosol mass concentrations corrected for the CDCE. We also highlight it in the methods section (lines 141-144):
**"LASIC employed an Aerodyne aerosol chemical speciation monitor (ACSM) to provide quantitative measurement of the chemical composition of non-refractory aerosol components including sulfate, nitrate, ammonium, and organics. For comparative analysis, we use aerosol concentrations corrected for composition-dependent collection efficiency (CDCE) obtained from the ARM Data Archive."**

Figure 2: It may help the reader to add a symbol indicating Ascension Island on this map and in the legend.

We added a symbol to the map in Figure 2 to indicate Ascension Island and referenced it in the caption as well.

Figure 4: how large are the standard deviations for each data point shown here (taken from monthly mean values across all sites)?

To clarify, we include standard deviation error bars for each data point in Fig. 4 and updated the text to reflect this change (lines 199-202):
**"Each data point corresponds to the monthly mean AOD values at distinct AERONET sites. The error bars in Fig. 4 represent the ±1 standard deviation in monthly AOD measurements at these sites, with higher deviations observed during the biomass burning months (up to ±0.25 at Namibe site)."**

Fig. A1: Are the modeled AOD averaged for grid points around Ascension Island or do they represent the entire domain? Please specify in the caption. If the entire domain was used, are you also able to show grid point values for near/around Ascension Island as this was the site used for comparison?

For consistency in our comparisons, we use the modeled AOD values at the location of the AERONET data collection site on Ascension Island. This approach was consistently applied across all AERONET sites discussed in the paper for AOD comparisons. We added it in our methods section (lines 129-131):

**"The modeled AOD is sampled at each AERONET site location and computed at 550 nm wavelength by vertically integrating scattering and absorption coefficients based on the properties of various aerosol components, such as size distributions, hygroscopicity, refractive indices, and densities (Latimer and Martin, 2019)."**

Line 202-203: Observational evidence of predominantly internally-mixed and aged biomass burning particles in the Southeast Atlantic from aircraft measurements has been shown by Dang et al. (2022), please include.

Thank you for bringing this to our attention – we added this citation in our paper.

Fig. A2 caption: nitrates in the figure legend are identified as "NO3" while in the caption nitrates are "NIT". Please correct for consistency.

Thank you for pointing it out—we made the correction as suggested.

Fig. A2: conventional ACSM/AMS composition coloring is typically, organics (green), nitrate (blue), sulfate (red), and ammonium (orange). For consistency with this convention and to not confuse the coloring scheme in Fig. A1, it is preferred that the authors stick with the ACSM/AMS convention.

We update the color scheme for Figures 5, 6, A1, and A2 to align with the conventional ACSM/AMS composition colors.

Figure 6: similar to Fig. A2 caption, please correct NIT and NO3 for consistency.

The caption has been corrected for consistency.

Line 325-326: Has previous work described or quantified the "substantial uncertainties in DMS concentrations…"? I feel as though a citation is needed here.

We add two citations, Asher et al. (2011) and Tortell et al., (2011), in the sentence to support the uncertainties observed in DMS concentrations:
**"The Benguela region has substantial uncertainties in DMS concentrations in surface seawater (Asher et al., 2011; Tortell et al., 2011) and the corresponding emission fluxes owing to the limited availability of biogenic sulfur measurements."**

Minor edits:

Line 47: "…, leading to [the] largest uncertainty of aerosol radiative forcing…"

This sentence has been rephrased in response to the comment of another reviewer as follows:
**"Although DMS is a critical source of natural aerosols, contributing over 50% of natural gas-phase sulfur emissions (Chin et al., 1996; Kilgour et al., 2021), the exact mechanisms of DMS oxidation and subsequent formation of sulfate and MSA remain inadequately understood (Ravishankara et al., 1997; Barnes et al., 2006; Hoffmann et al., 2016). This gap in understanding contributes to substantial uncertainty in aerosol radiative forcing, which is highly sensitive to uncertainties in natural aerosols (Carslaw et al., 2013; Fung et al., 2022)."**

Line 200: delete, "during" at the end of the sentence.

Thank you for noticing – this sentence has been omitted.

**Other changes:**

We found that coarse-mode sea-salt aerosols were inadvertently omitted from the model AOD calculations. We have now included them, which resulted in minor changes to Figures 3, 4, and A1.

**References added to the manuscript based on reviewer responses:**

Arowosegbe, O. O., Röösli, M., Adebayo-Ojo, T. C., Dalvie, M. A., and de Hoogh, K.: Spatial and Temporal Variations in PM10 Concentrations between 2010–2017 in South Africa, Int. J. Environ. Res. Public Health, 18, 1-12, doi.org/10.1016/j.envpol.2022.119883, 2021.

Asher, E.C., Merzouk, A., and Tortell, P.D.: Fine-scale spatial and temporal variability of surface water dimethylsufide (DMS) concentrations and sea–air fluxes in the NE Subarctic Pacific, Mar. Chem., 126, 63-75, https://doi.org/10.1016/j.marchem.2011.03.009, 2011.

Dang, C., Segal-Rozenhaimer, M., Che, H., Zhang, L., Formenti, P., Taylor, J., Dobracki, A., Purdue, S., Wong, P., Nenes, A., Iii, A., Coe, H., Redemann, J., Zuidema, P., Howell, S., and Haywood, J.: Biomass burning and marine aerosol processing over the southeast Atlantic Ocean: a TEM single-particle analysis, Atmospheric Chemistry and Physics, 22, 9389-9412, 10.5194/acp-22-9389-2022, 2022.

Fung, K. M., Heald, C. L., Kroll, J. H., Wang, S., Jo, D. S., Gettelman, A., Lu, Z., Liu, X., Zaveri, R. A., Apel, E. C., Blake, D. R., Jimenez, J.-L., Campuzano-Jost, P., Veres, P. R., Bates, T. S., Shilling, J. E., and Zawadowicz, M.: Exploring dimethyl sulfide (DMS) oxidation and implications for global aerosol radiative forcing, Atmos. Chem. Phys., 22, 1549–1573, https://doi.org/10.5194/acp-22-1549-2022, 2022.

Gong, S. L.: A Parameterization of Sea-salt Aerosol Source Function for Sub- and Super-micron Particles. Glob. Biogeochem. Cycles, 17(4), https://doi.org/10.1029/2003GB002079, 2003.

Kilgour, D. B., Novak, G. A., Sauer, J. S., Moore, A. N., Dinasquet, J., Amiri, S., Franklin, E. B., Mayer, K., Winter, M., Morris, C. K., Price, T., Malfatti, F., Crocker, D. R., Lee, C., Cappa, C. D., Goldstein, A. H., Prather, K. A., and Bertram, T. H.: Marine gas-phase sulfur emissions during an induced phytoplankton bloom, Atmos. Chem. Phys., 22, 1601–1613, https://doi.org/10.5194/acp-22-1601-2022, 2022.

Russell, L., Moore, R., Burrows, S., and Quinn, P.: Ocean flux of salt, sulfate, and organic components to atmospheric aerosol, Earth-Science Reviews, 239, 10.1016/j.earscirev.2023.104364, 2023.

Tortell, P.D., Guéguen, C., Long, M.C., Payne, C.D., Lee, P., and DiTullio, G.R.: Spatial variability and temporal dynamics of surface water pCO2, ΔO2/Ar and dimethylsulfide in the Ross Sea, Antarctica. Deep Sea Res. Part I Oceanogr. Res. Pap., 58, 241-259, https://doi.org/10.1016/j.dsr.2010.12.006, 2011.

Zhang, D., Du, L., Wang, W., Zhu, Q., Bi, J., Scovronick, N., Naidoo, M., Garland, R.M. and Liu, Y.: A machine learning model to estimate ambient PM2. 5 concentrations in industrialized highveld region of South Africa, Remote Sens. Environ., 266, 112713, 2021.